# ScanTD: 360° Scanpath Prediction based on Time-Series Diffusion

Yujia Wang
wang10@staff.vuw.ac.nz
Victoria University of Wellington
Wellington, New Zealand

Fang-Lue Zhang*
fanglue.zhang@vuw.ac.nz
Victoria University of Wellington
Wellington, New Zealand

Neil A. Dodgson
neil.dodgson@vuw.ac.nz
Victoria University of Wellington
Wellington, New Zealand

## Abstract

Scanpath generation in 360° images aims to model the realistic trajectories of gaze points that viewers follow when exploring panoramic environments. Existing methods for scanpath generation suffer from various limitations, including a lack of global attention to panoramic environments, insufficient diversity in generated scanpaths, and inadequate consideration of the temporal sequence of gaze points. To address these challenges, we propose a novel approach, named ScanTD, which employs a conditional Diffusion Model-based method to generate multiple scanpaths. Notably, a transformer-based time-series (TTS) module with a novel attention mechanism is integrated into ScanTD to capture the temporal dependency of gaze points effectively. Additionally, ScanTD utilizes a Vision Transformer-based method for image feature extraction, enabling better learning of scene semantic information. Experimental results demonstrate that our approach outperforms state-of-the-art methods across three datasets. We further demonstrate its generalizability by applying it to the 360° saliency detection task.

## CCS Concepts

• **Computing methodologies → Activity recognition and understanding**.

## Keywords

360° images; Scanpath; Diffusion Model; Vision Transformer

**ACM Reference Format:**
Yujia Wang, Fang-Lue Zhang, and Neil A. Dodgson. 2024. ScanTD: 360° Scanpath Prediction based on Time-Series Diffusion. In *Proceedings of the 32nd ACM International Conference on Multimedia (MM '24), October 28-November 1, 2024, Melbourne, VIC, Australia.* ACM, New York, NY, USA, 10 pages. https://doi.org/10.1145/3664647.3681315

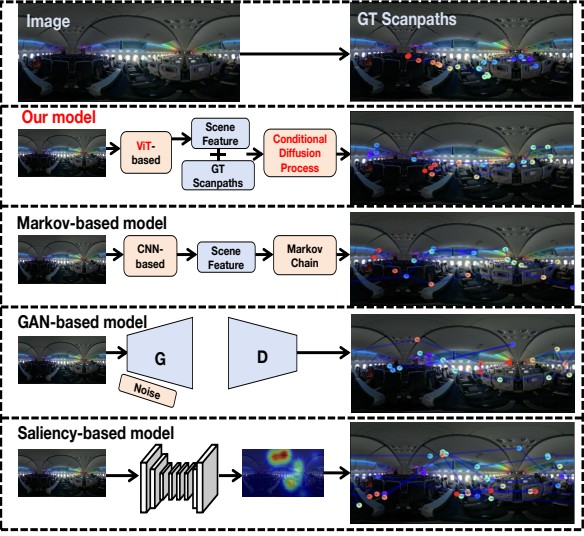

**Figure 1: Existing** 360° **scanpath prediction methods**

## 1 Introduction

In the rapidly evolving field of virtual reality (VR), understanding the visual attention of users is critical for creating immersive and interactive experiences. It offers meaningful insights into scene rendering, data transmission, and compression [11, 35] for developing better VR applications. However, recruiting actual participants

*Corresponding Author

and obtaining real experimental data using eye-tracking devices is time-consuming and complex. Therefore, it would significantly benefit this research field to develop a method capable of modeling human visual attention behavior and generating realistic visual attention patterns without the necessity for real-world experiments. 360° images are experiencing a surge in popularity due to their ability to deliver highly immersive viewing experiences. Generally, the study of human visual attention on 360° images has two research focuses: saliency maps and gaze point trajectories (scanpaths). Unlike saliency maps, which only present static results, gaze point movement in a scanpath is a dynamic process [5]. Recent studies have shown that the dynamics of gaze points can reflect various characteristics, such as the attractiveness and quality of areas viewed at different time stamps [38, 41, 43].

Scanpath generation aims to accurately model the trajectories of viewers' gaze points from two perspectives: the spatial positions of gaze points and their temporal order. The recent advances in deep learning techniques have enabled scanpath prediction in traditional 2D images [24, 25]. However, modeling scanpaths in 360° images presents a more challenging task due to the significant differences from 2D planar images. In 360° environments, observers engage not only in scanning the image with their gazes but also in changing their viewports by turning their heads or bodies [43]. This also necessitates better global scene feature extraction in the larger space offered by 360° images. Additionally, most studies in scanpath modeling do not fully address the temporal dynamics of gaze points, which are crucial for accurately simulating realistic gaze point trajectories [5]. Capturing this temporal dependency among gaze points is particularly vital in interactive applications such as VR training simulation and interactive storytelling.

Existing research and human attention datasets also indicate that, while observers may share similar gaze patterns for the same scene, there remains significant diversity in gaze behavior during free-viewing [32, 35, 42]. Therefore, generating multiple scanpaths is critical for real-world applications. Researchers [2, 27] have employed Generative Adversarial Networks (GANs) to generate diverse scanpaths (see Fig. 1). However, GAN-based methods suffer from instability issues, such as mode collapse. An alternative approach is Markov-based methods [12, 37] that use a Markov Chain to predict scanpaths (see Fig. 1), but they struggle with handling long-range dependencies due to their reliance on the immediate last state, leading to the loss of relevant information over time.

In this research, we propose a novel approach, ScanTD, to address the above challenges in 360° scanpath prediction. Based on recent research that confirmed the better capability of the Diffusion Model in learning to generate diverse sequential outputs [16, 21], we incorporate the Diffusion Model into the scanpath prediction task. We employ a conditional Diffusion Model that uses the deep features of the given 360° image as its condition and effectively learns the distribution of scanpath data via a gradual denoising procedure. Previous studies model the temporal domain information based on Recurrent Neural Networks (RNNs) [19, 30] or Markov Models [37], which struggle to cope with increasing lengths of time series and suffer from low computational efficiency. Recently, Transformer-based time-series processing models [48] have demonstrated superior performance in capturing long-range temporal dependencies, achieving higher efficiency than RNN-based methods. In our conditional diffusion model, we adopt a Transformer-based time-series (TTS) module in the denoising procedure to capture the temporal information across time-ordered gaze points.

When extracting 360° image features for our diffusion model, we take into account the spherical nature of 360° images to address the typical distortions [13] to enhance the scanpath learning process. Most existing scanpath modeling methods for 360° images rely on normal 2D CNNs, which often struggle to capture distant spatial relationships between content elements [1, 27, 37]. Instead, we utilize Vision Transformer [17], incorporated with a Spherical CNN [9], to learn the global scene features in 360° images. This use of global attention significantly enhances the accuracy and realism of scanpath modeling in 360° images.

Our main contributions include: (1) We propose a novel 360° scanpath prediction method based on time-series diffusion, ScanTD, which boosts the performance of scanpath generation in 360° images. (2) We develop a transformer-based time-series (TTS) module with a novel temporal attention mechanism to better capture the dynamic temporal information across time-ordered gaze points. This modification significantly improves the accuracy of both the spatial positions and the temporal order of the generated gaze points. (3) We integrate Spherical CNNs into Vision Transformer for better global scene feature extraction in 360° images.

## 2 Related Works

### 2.1 Scanpath Modeling on 2D/360° Images

Scanpath prediction and generation have been explored in traditional 2D images for many years and have achieved notable progress. Scanpaths differ from saliency maps, in that they are time-aware.

These time-aware models [5] highlight the limitations of static saliency maps by underscoring the importance of incorporating temporal dynamics into the modeling of eye movement trajectories. In recent years, the application of Deep Learning and Probabilistic Generative Models [14, 22, 25] has proven the efficacy of neural networks in recognizing and understanding the complex patterns of human visual attention. However, the performance of these models can be sensitive to certain parameters, such as the number of Gaussian components, which limits their ability to handle complex scenes. Markov chain-based methods [12] for gaze behavior modeling provide insights into integrating various influences on gaze behavior, yet they fall short in efficiently capturing the temporal dependencies in gaze points and shifts in visual attention. Beyond scanpath modeling for free-viewing situations, several studies [8, 45] have also explored goal-directed human visual attention modeling.

For 360° images, several attempts have been made to model saliency maps [6, 28, 29, 35, 36], but scanpath modeling has received less attention. Initially, several scanpath prediction models based on saliency maps were developed, but their performance largely depends on the accuracy of the saliency map predictions [1, 3, 50, 51] (see Fig. 1). While some methods have focused on improving saliency prediction models and sampling strategies, constructing effective sampling strategies for time-dependent visual behavior remains challenging, as evidenced by the unstable results from models like SaltiNet [1]. Assens et al. [2], have attempted to adapt 2D models for 360° images but have encountered difficulties in accurately replicating ground truth scanpaths behaviors. Approaches using generative models, such as GANs, all struggle with training stability [2, 16]. The recent ScanDMM model [37], which simulates visual working memory through a Markov chain, shows promise in predicting realistic scanpaths. However, it struggles with the loss of state information over longer timescales, a limitation inherent to the Markov chain model itself. Furthermore, while ScanDMM [37] offers a probabilistic generative approach to capture gaze behavior diversity, it is constrained by the Markov assumption that future states depend solely on the current state, potentially limiting scanpath diversity due to this sequential dependence. This approach effectively models individual scanpaths but falls short in capturing the diversity of scanpaths across different viewers. In this context, our model, ScanTD, allows for more flexible modeling of temporal dependencies and panoramic scene semantic features, generating more diverse and realistic scanpaths. It is worth noting that all these studies use CNN-based methods for scene feature extraction in 360° environments, which are limited by their local perception. Moreover, employing CNN-based methods without pre-trained models for scene feature extraction in 360° images may result in poor generalization and challenges in capturing the complex spatial relationships inherent in panoramic content. In contrast, pre-trained models like the Vision Transformer (ViT) [17] offer robustness and enhanced generalization across diverse visual content, potentially improving scanpaths generation accuracy.

### 2.2 Diffusion Model

Diffusion models have shown success in various visual applications, including image production[33, 49]and audio synthesis[26]. These models consist of two processes: forward and reverse. The forward

process adds Gaussian noise to the original data distribution $x_0 \sim q(x_0)$, generating latent variables $x_1$ through $x_T$:

$$q(x_t \mid x_{t-1}) = \mathcal{N}(x_t; \sqrt{1 - \beta_t} x_{t-1}, \beta_t I) \quad (1)$$

The reverse process aims to reconstruct the original data using a learned neural network:

$$p_\theta(x_{t-1} \mid x_t) = \mathcal{N}(x_{t-1}; \mu_\theta(x_t, t), \Sigma_\theta(x_t, t)) \quad (2)$$

Here, $\beta_t \in (0, 1)$ is the noise variance at time $t$, and $\mu_\theta$ and $\Sigma_\theta$ are learned parameters. Diffusion probabilistic models excel in matching complex data distributions and managing long-range dependencies by reversing a gradual, multi-step noising process [21]. Dhariwal et al. [16] highlight their advantages over GANs in synthesizing high-quality images, with a structured generative process providing greater stability and robustness. Additionally, diffusion models outperform GANs in covering distribution modes, suggesting superior capability in handling data variability [31]. For scanpath modeling in 360° images, the unique demands align well with the strengths of diffusion models, enabling them to navigate complex spatial relationships and temporal sequences effectively. Similar to conditional-VAE and conditional-GAN, diffusion models can treat the encoded latent input, $X$, as a condition [4, 47]. This approach extends to time-series prediction and audio waveform generation in models like CSDI [39] and WaveGard [7]. Traditionally, diffusion models for image-to-image processing use a U-net architecture [34]. However, for sequence-to-sequence tasks like text generation, Transformer-based methods have been employed as alternatives [20]. For instance, CSDI [39] uses a Transformer-based model with two-dimensional attention mechanisms for capturing temporal and feature dependencies in multivariate time-series data. Inspired by these advances, our proposal, ScanTD, leverages scene features from 360° images as conditions to generate varied scanpaths. Specifically, we introduce a transformer-based time-series module with a novel multi-attention mechanism, replacing the U-net to learn the temporal order of gaze points more effectively. This innovation allows ScanTD to better capture the temporal and feature dependencies in multivariate time-series data, making it particularly suited for scanpath modeling.

## 3 Method

**Problem statement.** In a 360° environment, viewers' gaze point movements can be considered as a dynamic, temporally ordered process. A human scanpath can be defined as a time series of gaze points $X_{1:T} = (X_1, X_2, \ldots, X_T) \in \mathbb{R}^{3 \times T}$, where $X_t$ is the 3D coordinate $(x_t, y_t, z_t)$ of a gaze point. Specifically, given a 360° image, the scanpath prediction method aims to generate realistic scanpaths $\hat{X}_{1:T}$. For better real-world application and to simulate different viewers, the method needs to efficiently generate multiple scanpaths $\hat{X}_1, \hat{X}_2 \ldots \hat{X}_m$ for one given image, ensuring the diversity of the generation results [35].

## 3.1 Overview

As illustrated in Fig. 2, ScanTD comprises two networks. The first network is used for scene feature extraction from 360° images, using a Vision Transformer-based method. The second network is a diffusion model-based network that is conditioned on the extracted scene features to generate various scanpaths. Specially, it involves

a Transformer-based time-series (TTS) module when denoising to deal with the temporal sequence data.

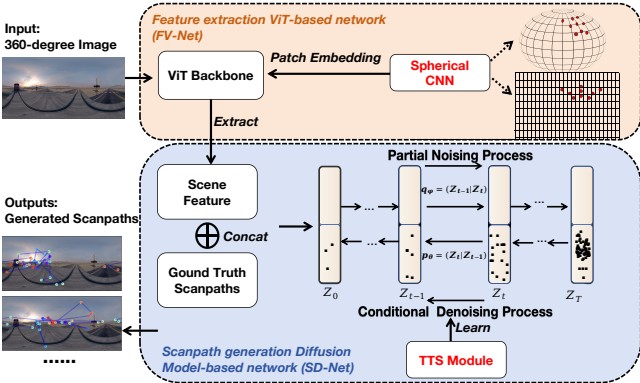

**Figure 2: The learning framework of ScanTD**

**Feature Extraction ViT-based Network (FV-Net).** In the FV-Net, we employ Vision Transformer (ViT) for its superior global perception capabilities in 360-degree scene feature extraction. However, given that ViT models are originally designed for conventional 2D images, their performance is constrained by the inherent distortions in 360° images. To address this limitation, we substitute the standard 2D patch embedding in ViT with Spherical CNN [10]. This adaptation significantly enhances the extraction of the features of the omnidirectional visual content in the input 360° image.

**Scanpath Generation Diffusion Model-based Network (SD-Net).** In the SD-Net, we introduce a novel conditional diffusion model to generate diverse scanpaths, where the extracted scene features are used as the condition. The SD-Net facilitates a more context-aware generation of scanpaths. More specifically, to effectively cope with the dynamic temporal sequence of gaze points in scanpaths, we replace the conventional U-net in the original Diffusion Model with a Transformer-based time-series (TTS) module for modeling the diffusion process on the sequential data.

## 3.2 FV-Net

Given a 360° image, the output of this stage is an image feature embedding with a shape of $B \times 20 \times 196$ ($B$ is the batch size). Different from traditional CNN-based methods, ViT is a Transformer-based architecture, devoid of convolutions to process flattened image patches, enabling it to effectively capture long-range dependencies between different regions of an image. This convolution-free approach of the ViT not only challenges the prevailing CNN paradigms but also sets new benchmarks in large-scale image classification tasks, achieving state-of-the-art performance on datasets like ImageNet [15]. The shape of vanilla ViT [17] is $B \times 197 \times 768$, and we remove the last layer used for image classification and use the remaining ones with a shape of $B \times 196 \times 768$ to output the scene features of the image.

Traditional 2D CNNs can not handle the distortion introduced by the equirectangular projection-based representation of 360° images. Therefore, we replace 2D CNNs with Spherical Convolutions [10] that have been shown to perform better for embedding equirectangular content in the ViT model. Spherical convolutions are a

type of dilated convolution where the relationships between image elements are not established in 2D image space, but in a gnomonic, non-distorted spherical space. These spherical convolutions can represent kernels as patches tangent to a sphere where the 360° is reprojected, and therefore allow the network to learn spatial relations of 360° content, such as longitudinal continuities or spherical distortions. The core of Spherical Convolutions is adapting convolutional neural networks for 360° image data by employing filter kernels that have a perception field covering the pixel positions aligned with the spherical nature of the images, effectively handling the distortions typical in 360° data [10]. In a traditional ViT model, a 2D image $I$ is divided into $N$ 2D image patches $P_1, P_2, \ldots, P_N$. Each image patch $P_i$ is transformed into a one-dimensional vector $v_i$ through a patch embedding layer $E$: $v_i = E(P_i)$. Then, the vectors $v_1, v_2, \ldots, v_N$ are fed into the Transformer encoder. In our modified approach, we replace the 2D patch embedding with spherical CNNs to extract features from the input 360° image, preserving its spherical properties. These features are then partitioned into patches, each shaped into a feature vector. The resulting feature vector $F$ has a shape of $B \times 196 \times 768$ and is subsequently fed into the ViT architecture.

## 3.3 SD-Net

We use a conditional time-series data diffusion model for learning to generate scanpaths with better diversity and accuracy based on the extracted scene features. As shown in Fig. 2, our model learns to denoise the gaze point sequence to make it progressively closer to the real data distribution through denoising steps. We use the scene features as the condition of the corresponding scanpaths of the same image by concatenating the features and the point sequence to form the sample space to model by the diffusion model. Notably, the noise is not added to the conditional scene features in the diffusion process. A Transformer-based time-series module is adopted to denoise the scanpath with the given condition, which enables a better exploration of the temporal dependencies among gaze points within the same scanpath.

**Scene-Conditioned Scanpath Partial Noising**

We denote the scene feature of an image as $\mathbf{C}$ with a shape of $B \times 20 \times 196$ and denote the corresponding ground truth scanpaths as $\{\mathbf{S}^p\} = \{\mathbf{S}^1, \mathbf{S}^2 \ldots \mathbf{S}^n\}$ ($n$ is the number of ground truth scanpaths) of that image. Each scanpath is a $20 \times 3$ matrix, representing a temporal sequence of 20 gaze points. We use the 3D coordinates of the point on the unit sphere $x, y, z$ to represent each point. Our conditional diffusion model learns to model the sample space where each sample is a combination of the image scene information $\mathbf{C}$ and one of the ground truth scanpaths $\mathbf{S}^p$, expressed as $\mathbf{z} = \mathbf{C} \oplus \mathbf{S}^i = [C, S_1^p, S_2^p \ldots S_m^p]$, where $m$ is the point number of a scanpath, normally set as 20. Note that if an image has $n$ ground truth scanpaths, there will be $n$ corresponding samples in the dataset used for learning. The forward diffusion process is expressed as:

$$q_\phi(\mathbf{z}_t | \mathbf{z}_{t-1}) = \mathcal{N}(\mathbf{z}_t; \sqrt{1 - \beta_t} \mathbf{z}_{t-1}, \beta_t I). \tag{3}$$

Here, $\beta_t$ is a small constant that controls the noise level. $t$ represent the state of the variable $\mathbf{z}$ during the process. Different from conventional diffusion models that corrupt the whole $\mathbf{z}_t$ (both image scene

information $\mathbf{C}$ and a ground truth scanpath $\mathbf{S}_t^p$), we only add noises to the different states of the scanpath to obtain $\mathbf{S}^{p,1}, \mathbf{S}^{p,2}, \ldots, \mathbf{S}^{p,T}$, where $\mathbf{S}^{p,t}$ represents the $t$-th state of the scanpath $\mathbf{S}^p$ during the partial noising process. The conditional part $\mathbf{C}$ remains unchanged during the forward process, making it a partial noising process. This modification allows for the generated scanpaths that not only mimic the variability found in human visual attention but also are deeply informed by the scene's content structure. In practical applications, this translates to the ability to create highly personalized user experiences in VR settings. By understanding the scene cues that guide visual attention, content creators can design experiences that are both informative and visually captivating, leading to enhanced engagement and satisfaction.

**TTS-based Conditional Denoising**

We recover the original scanpath $\mathbf{S}^p$ in the reverse process using a conditional denoising module. Particularly, we utilize a Transformer-based time-series (TTS) learning module, denoted as $f_\theta$, to learn to reconstruct the original $\mathbf{z}_0$ given the conditional part $\mathbf{C}$ that represents the scene's visual features.

We adopt the TTS module to learn the transition from a later noised ground truth state $\mathbf{S}^{p,t}$ back to an earlier noised state $\mathbf{S}^{p,t-1}$, thereby effectively recovering the sequence of gaze points. Our TTS module comprises eight 2D multi-head self-attention blocks. As illustrated in Fig. 3, each block integrates a spatial multi-head attention alongside a temporal multi-head attention, enabling the extraction of hidden relationships across both temporal and spatial dimensions. The spatial attention operates on the tensor of each feature to learn spatial dependencies, whereas the temporal attention processes the tensor of each timestamp of the gaze points to discern temporal dependencies. The input embedding with shape $[B, L, C]$ (batch size, length, channel number) is initially expanded in dimension to $[B, L, C, D]$ (batch size, length, channel number, depth). Within each 2D Multi-head Self-attention block, the expanded embedding undergoes a split by channel to facilitate feature extraction via spatial attention, followed by a sequence split for temporal feature extraction through temporal attention.

Crucially, for the TTS module, we use a two-dimensional attention mechanism in each residual layer instead of convolution to capture spatial and temporal dependencies of multivariate time series. We employ a spatial Transformer layer and a temporal Transformer layer, both as 1-layer Transformer encoders. The spatial Transformer layer takes the features of each spatial position as inputs to learn spatial dependencies, whereas the temporal Transformer layer processes tensors at each time point to learn temporal dependencies. This capability is important for real-world applications, predicting where and when a viewer might look next can guide the adaptive streaming of high-definition content to just the right places at the right times.

In the reverse process, we use the TTS module $f_\theta$ to estimate the posterior distribution:

$$p_\theta(z_{0:T}) := p(\mathbf{z}_T) \prod_{t=1}^{T} p_\theta(\mathbf{z}_{t-1} | \mathbf{z}_t, \mathbf{z}_0), \quad \mathbf{z}_T \sim \mathcal{N}(0, \mathbf{I}) \tag{4}$$

$$p_\theta(\mathbf{z}_{t-1} | \mathbf{z}_t, \mathbf{z}_0) = \mathcal{N}(\mathbf{z}_{t-1}; \mu_\theta(\mathbf{z}_t, t | \mathbf{z}_0), \sigma_\theta(\mathbf{z}_t, t | \mathbf{z}_0) \mathbf{I}) \tag{5}$$

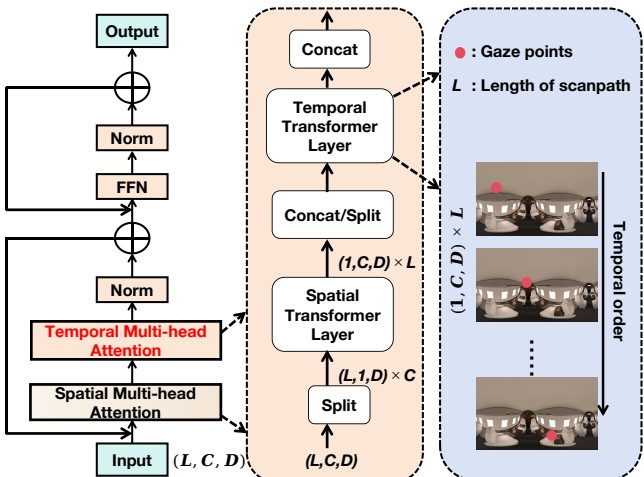

**Figure 3: The architecture of the TTS module.**

where the $\mu_\theta$ and $\sigma_\theta$ represent the parameters for the predicted mean and standard deviation of $q_(\mathbf{z}_t - 1 | \mathbf{z}_t, \mathbf{z}_0))$ from the forward nosing process. The condition $\mathbf{C}$ is included in $\mathbf{z}$ and remains unchanged. After the conditional denoising process from $t = T$ to $t = 0$, we generate $\mathbf{z}_{0:T} = \mathbf{C} \oplus \hat{\mathbf{S}}^{p,0:T}$.

During the training process of our TTS-based denoising model, we optimize the parameters of the TTS module by the error between the generated scanpath states $\hat{\mathbf{S}}^{1:T}$ and the ground truth scanpaths $\mathbf{S}^{1:T}$. The losses are computed following the original denoising diffusion model proposed in [21]:

$$
\mathcal{L}_{VLB} = \mathbb{E}_{q(\mathbf{z}_{1:T}|\mathbf{z}_0)} \Bigg[ \underbrace{\log \frac{q(\mathbf{z}_T|\mathbf{z}_0)}{p_\theta(\mathbf{z}_T)}}_{L_T} + \tag{6}
$$

$$
\underbrace{\sum_{t=2}^{T} \log \frac{q(\mathbf{z}_{t-1}|\mathbf{z}_0,\mathbf{z}_t)}{p_\theta(\mathbf{z}_{t-1}|\mathbf{z}_t)}}_{L_{t-1}} \underbrace{- \log p_\theta(\mathbf{z}_0|\mathbf{z}_1)}_{L_0} \Bigg]
$$

The reverse process can be simplified by solving the following optimization problem:

$$
\min_\theta \mathcal{L}_{\text{VLB}} = \min_\theta \left[ \sum_{t=1}^{T} \|\mathbf{z}_0 - f_\theta(\mathbf{z}_t, t)\|^2 \right]
$$
$$
\rightarrow \min_\theta \left[ \sum_{t=1}^{T} \|\hat{\mathbf{S}}^{p,0} - \tilde{f}_\theta(\mathbf{z}_t, t)\|^2 \right] \tag{7}
$$

where the $\tilde{f}_\theta(\mathbf{z}_t, t)$ donates the fractions of reconstructed $\mathbf{z}_0$ corresponding to $\hat{\mathbf{S}}^{p,0}$.

### 3.4 Scanpath Generation

To predict scanpaths from a given 360° image, **FV-Net** extract global scene features $\mathbf{C}$, which serve as the condition for the following SD-Net. Subsequently, a standard noise $\mathbf{N}$ with the shape of $(1, 20, 3)$, mirroring the shape of ground truth scanpaths, is generated based

on the standard normal distribution. This noise is then concatenated with the condition $\mathbf{C}$ obtained from the **FV-Net** as: $\mathbf{W} = \mathbf{C} \oplus \mathbf{N}$. The concatenated tensor is subsequently fed to **SD-Net**. Within **SD-Net**, the mean and variance of the concatenated tensor $\mathbf{W} = \mathbf{C} \oplus \mathbf{N}$ are used to denoise. This denoising process is iterated until a preset number of noising/denoising cycles are completed, resulting in the concatenation of the condition and the generated scanpaths: $\mathbf{C} \oplus \hat{\mathbf{S}}^p$. To ensure that the generated scanpaths exhibit the necessary diversity reflective of natural human gaze behavior when viewing images in a 360° environment, we utilize different noise samples to generate scanpaths for the same image.

## 4 Experiments and Results

### 4.1 Datasets

We use three 360° image datasets: Sitzmann [35], AOI [42], and Salient360! [32]. Sitzmann [35] comprises 22 360° images and 1,980 scanpaths collected from 169 different users. Each scanpath represents gaze information recorded over 30 seconds at a sampling rate of 120Hz. To simplify the data, we reduced the sampling frequency of these scanpaths to 1Hz, yielding 30 data points per scanpath. To address the limited size of the dataset, we augmented the data by rotating the 360° images longitudinally and making corresponding adjustments to the scanpaths, thereby creating six distinct versions for each image. We used 19 of the 22 images for training, and the remaining three were included in the test set, effectively expanding our training dataset to 114 images. The AOI dataset [42] comprises 600 high-resolution 360° images accompanied by 18,000 scanpaths, and we randomly select 100 360° images for model evaluation. We use the training set in Salient360! [32], which contains 85 images and 3,036 scanpaths, for evaluation as well.

### 4.2 Experimental Setup

**Evaluation Metrics.** Many metrics have been proposed to quantitatively evaluate the similarity between two disparate scanpaths [18]. We used three metrics to quantitatively evaluate the performance of our ScanTD model: Levenshtein distance (LEV) , Dynamic Time Warping (DTW) , and Recurrence measure (REC) [18, 27]. LEV quantifies the similarity between two strings. In practical applications, LEV can also be employed to compare ordered sequences. In our experiment, the scanpath is defined as a series of 3D coordinate points $X_{1:T} = (X_1, X_2, \ldots, X_T) \in \mathbb{R}^{3 \times T}$. We convert these coordinate points into strings and then use LEV to assess their similarities. LEV is computed using the dynamic programming formula:

$$
\text{LEV}(A, B) = \min \Big( \text{LEV}(A - 1, B) + 1,
$$
$$
\text{LEV}(A, B - 1) + 1, \tag{8}
$$
$$
\text{LEV}(A - 1, B - 1) + \delta_{A[A] \neq B[B]} \Big)
$$

where $\delta$ is the Kronecker delta function that equals 0 if $A[A] = B[B]$ and 1 otherwise. DTW is a metric designed to compare the similarity of different time-series sequences. The gaze points in a scanpath exhibit temporal order characteristics, and the output coordinate points typically contain $(x_t, y_t, z_t)$, with each point representing a time step. Therefore, DTW can be used to measure the ability of each method in capturing the dynamic temporal order dependencies

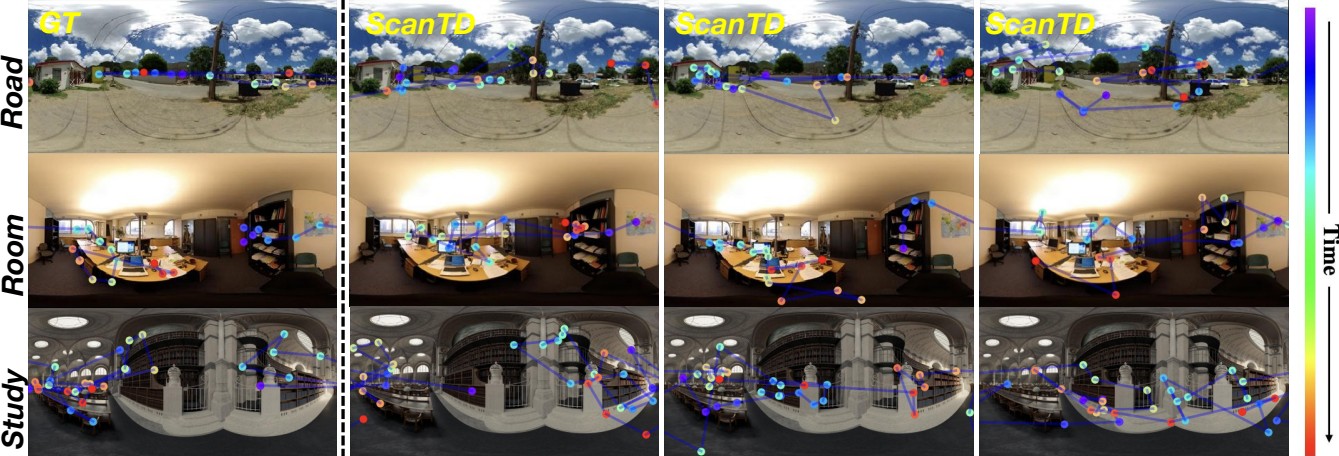

**Figure 4: Diverse generation results of ScanTD on three datasets**

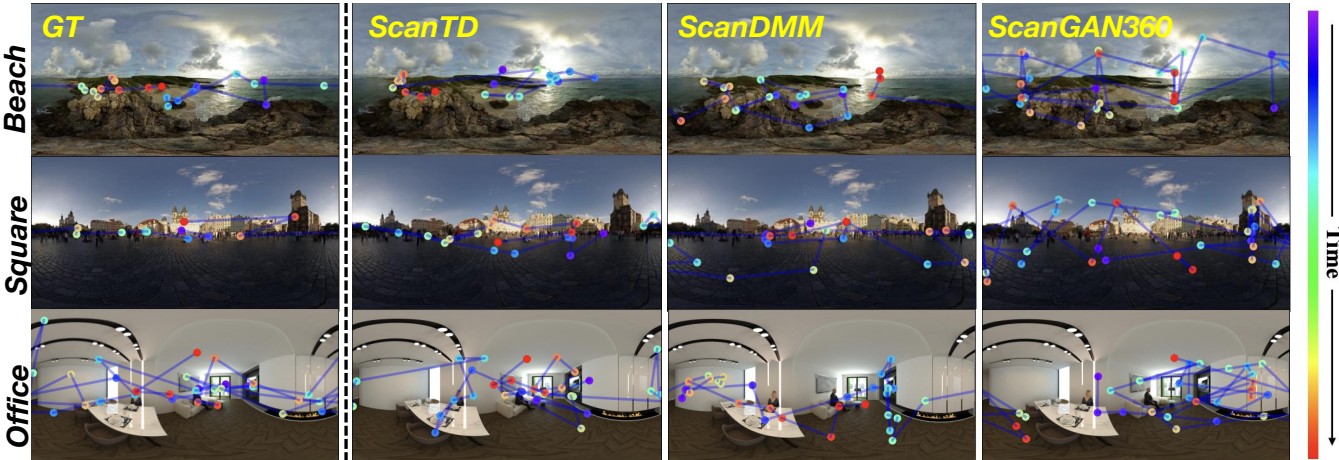

**Figure 5: Qualitative comparison to different scanpath prediction models on three datasets**

of scanpath. It is computed as:

$$\text{DTW}(A, B) = \sqrt{\sum_{i=1}^{n} \sum_{j=1}^{m} w(i, j) \cdot (\text{dist}(A[i], B[j]))^2} \quad (9)$$

where $\text{dist}(A[i], B[j])$ is the distance between points $A[i]$ and $B[j]$, and $w(i, j)$ is a weight function. Recurrence Measure (REC) is normally used to study the recurrence properties of time series data. Given a time series $A$ with length $n$, REC is calculated as the proportion of recurrence points $(R)$ by:

$$\text{REC}(A) = \frac{1}{n^2} \sum_{i=1}^{n} \sum_{j=1}^{n} \Theta(\varepsilon - \|A[i] - A[j]\|) \quad (10)$$

where $\Theta$ is the Heaviside step function, $\varepsilon$ is the threshold distance, and $\|\cdot\|$ denotes the Euclidean distance. Lower LEV/DTW values and higher REC values indicate better performance.

**Training Details and Inference Time.** FV-Net and SD-Net are trained simultaneously. We implement these two networks by Py-Torch and train them for 500 epochs. The resolution of training images is $448 \times 224$, and in each training iteration, the batch size is 8. The parameters of these two networks are both updated using the Stochastic Gradient Descent optimizer. The initial learning rate for these two networks is set to $1 \times 10^{-3}$; the weight decays are both applied with $5 \times 10^{-5}$ for regularization. The timestep in SD-Net is 500. It takes 0.28 seconds to generate one scanpath and 4.73 minutes to generate 1000 scanpaths using an NVIDIA RTX 3090 Ti GPU.

### 4.3 Diversity and Performance Comparison

**Diverse Results of ScanTD.** As illustrated in Fig. 4, our approach ScanTD is capable of generating multiple scanpaths for the same scene, catering to the diversity required in real-world applications. Moreover, it ensures that the generated scanpaths are plausible and focus on relevant areas of interest within the scene. The road scene is from AOI dataset [42], the room scene is from Salient360! dataset [32], and the study scene is from Sitzmann dataset [35].

**Selection of Methods for Comparison.** We compare our approach ScanTD with three state-of-the-art 360° scanpath prediction models: SaltiNet [1], ScanGAN360 [27], ScanDMM [37] and

Table 1: Quantitative comparison of methods on three datasets

| Dataset | Method | LEV ↓ mean / best | DTW ↓ mean / best | REC ↑ mean / best |
|---|---|---|---|---|
| Sitzmann[35] | Random walk | 62.91 / 57.45 | 3011.78 / 2310.10 | 1.42 / 2.77 |
| | SaltiNet [1] | 50.04 / 49.83 | 2102.70 / 2065.49 | 1.96 / 2.21 |
| | TAHA [44] | 51.11 / 49.04 | 2104.21 / 2072.18 | 2.13 / 2.52 |
| | HAT [46] | 48.24 / 47.97 | 2101.20 / 2045.75 | 2.74 / 2.93 |
| | ScanGAN360 [27] | 45.16 / 43.98 | 1962.27 / 1951.13 | 1.96 / 2.15 |
| | ScanDMM [37] | 43.09 / 42.25 | 1932.62 / 1927.41 | 3.54 / 3.58 |
| | **ScanTD (ours)** | **42.91 / 41.07** | **1894.26 / 1891.31** | **3.97 / 4.12** |
| | Human | 39.87 / 38.54 | 1847.38 / 1828.98 | 7.92 / 6.94 |
| AOI [42] | Random walk | 16.23 / 13.87 | 849.25 / 743.57 | 2.09 / 2.85 |
| | SaltiNet [1] | 14.46 / 13.96 | 604.33 / 601.46 | 2.34 / 2.28 |
| | TAHA [44] | 17.84 / 17.02 | 597.36 / 594.51 | 2.42 / 2.66 |
| | HAT [46] | 13.21 / 13.09 | 562.39 / 658.13 | 2.65 / 2.71 |
| | ScanGAN360 [27] | 12.87 / 12.41 | 554.73 / 551.16 | 3.76 / 3.89 |
| | ScanDMM [37] | 12.32 / 12.10 | 529.08 / 520.97 | 4.27 / 4.36 |
| | **ScanTD (ours)** | **12.18 / 12.09** | **497.44 / 459.65** | **4.76 / 4.89** |
| | Human | 10.23 / 8.70 | 395.08 / 378.39 | 6.94 / 8.01 |
| Salient360! [32] | Random walk | 48.19 / 43.78 | 2417.40 / 2342.59 | 1.75 / 2.68 |
| | SaltiNet [1] | 41.31 / 40.95 | 1842.39 / 1841.07 | 2.47 / 2.51 |
| | TAHA [44] | 45.62 / 44.20 | 1812.34 / 1809.38 | 3.02 / 3.25 |
| | HAT [46] | 44.69 / 42.90 | 1801.30 / 1795.12 | 3.47 / 3.53 |
| | ScanGAN360 [27] | 39.01 / 38.72 | 1720.51 / 1706.10 | 3.58 / 3.72 |
| | ScanDMM [37] | 37.85 / 37.02 | 1519.06 / 1507.62 | 3.68 / 3.73 |
| | **ScanTD (ours)** | **37.67 / 37.20** | **1484.32 / 1470.51** | **3.98 / 4.07** |
| | Human | 34.17 / 32.98 | 1474.22 / 1397.76 | 4.75 / 5.92 |

two state-of-the-art 2D scanpath prediction methods that support free-view mode, including HAT[46] and TAHA[44]. In addition, we noted a newly posted arXiv paper that also uses the Diffusion Model for scanpath prediction [23]. However, it did not share its source code and trained models. Furthermore, its evaluation metrics differ from those of the published related works ScanDMM [37] and ScanGAN360 [27], and it did not provide sufficient details of their evaluation. Therefore, we chose not to compare with [23].

**Our Comparison Methodology.** ScanTD can produce any number of scanpaths as required to simulate different viewers. To perform a fair comparison, we use each method (SalitiNet [1], ScanDMM [37], ScanGAN360 [27], HAT [46], TAHA [44] and our ScanTD) to generate multiple scanpaths for each image and calculate both the average and the best results for each method using the evaluation metrics. Specifically, we compare each predicted scanpath against each ground truth scanpath, averaging the results for every image. Finally, the average result across each dataset is calculated. We also select the scanpath that best matches a corresponding ground truth scanpath for each image and then calculate the average of these results across the entire dataset. To minimize potential bias from randomly generated scanpaths, we tested each model 10 times per image. The results were then averaged to determine the final performance metrics as in [37]. For enhanced comparability and interpretability of the results, we calculate the human consistency [27] for each metric, serving as a realistic upper limit for model performance (refer to "Human" in Table 1). Additionally, we contrast our results with those of a chance model, which generates

scanpaths by incorporating random Brownian motion into prior positions (refer to "Random walk" in Table 1) [37].

**Comparison Results Analysis.** Table 1 and Figure 5 show that our method, ScanTD, can generate results closer to the ground truth, with predicted gaze points showing reasonable displacements in the vertical direction. The beach scene is from the Salient360! dataset [32], and the temporal order of our predicted gaze points across different parts of the scene more closely matches the ground truth compared to other methods. The square scene is from the AOI dataset [42] and our predicted gaze points concentrated on significant buildings. The office scene is from Sitzmann [35], and the color distribution of gaze points demonstrates that ScanTD can better capture the temporal order of gaze points.

### 4.4 Ablation Study

We conducted ablation experiments to analyze the role of Spherical CNNs [9] in scene feature extraction and the ability of our designed TTS module to capture temporal information of gaze points. We compare ScanTD with three baselines: (1) $ScanTD_1$, using original ViT without spherical CNNs [9] for 360° image scene feature extraction; (2) $ScanTD_2$, using Diffusion Model with original transformer without TTS module for scanpath generation; (3) $ScanTD_3$, performing both of these two modifications. All baseline models are trained in the same settings, and the results are shown in Table 2. We observe that the model achieves the best performance when incorporating both the Spherical CNNs [9] and our designed

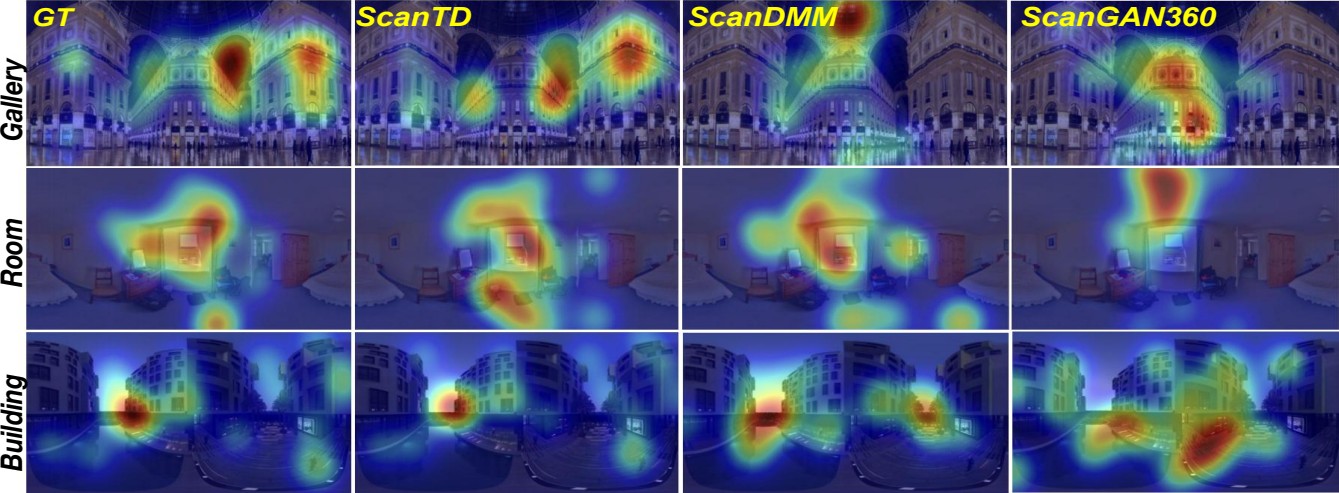

**Figure 6: Qualitative comparison to different saliency detection models on three datasets**

**Table 2: Ablation Study about the effectiveness of our design.**

| Database | Method | LEV ↓ | DTW ↓ | REC ↑ |
|---|---|---|---|---|
| Sitzmann[35] | $ScanTD_1$ | 46.80 | 1996.45 | 3.09 |
| | $ScanTD_2$ | 45.94 | 2098.41 | 2.83 |
| | $ScanTD_3$ | 47.48 | 2111.15 | 2.60 |
| | **ScanTD** | **42.79** | **1899.70** | **3.98** |
| | $ScanTD_4$ | 42.93 | 1941.20 | 3.37 |
| | $ScanTD_5$ | 42.91 | 1941.18 | 3.86 |
| Salient360![32] | $ScanTD_1$ | 39.19 | 1698.517 | 3.168 |
| | $ScanTD_2$ | 38.74 | 1787.67 | 2.74 |
| | $ScanTD_3$ | 39.98 | 1874.89 | 2.42 |
| | **ScanTD** | **38.12** | **1404.32** | **4.17** |
| | $ScanTD_4$ | 38.17 | 1507.09 | 3.95 |
| | $ScanTD_5$ | 37.88 | 1499.73 | 3.91 |

TTS module. Moreover, to determine whether the improved performance is due to the employed TTS modules rather than the larger number of parameters, we employed ten spatial transformer blocks in $ScanTD_4$ to ensure a similar number of parameters with the original $ScanTD$ that has eight TTS blocks; both models have sizes of approximately 230 MB. Additionally, we compared another model, $ScanTD_5$, in which the number of TTS module blocks is reduced to four, to explore the impact of the number of parameters on model performance.

### 4.5 Generalizability Analysis

We demonstrate the generalizability of our approach by applying it to saliency detection in 360° images. Saliency detection in 360° images aims to identify and predict areas within panoramic images that are most likely to capture and hold the viewer's visual attention. Intuitively, the series of gaze points generated by a scanpath prediction approach should be able to used to indicate the regions where the users concentrate within an image. We use the scanpaths predicted by ScanGAN360 [27], ScanDMM [37] and our method to

generate saliency maps. Following the method introduced in [37] to convolve fixation maps, we use an adapted Gaussian function [40] to generate a saliency map from multiple predicted scanpaths:

$$G(x,y) = \frac{1}{2\pi\sigma_y^2} \exp\left(-\frac{x^2}{2\sigma_x}\right) \exp\left(-\frac{y^2}{2\sigma_y}\right) \tag{11}$$

where $\sigma_x = \frac{\sigma_y}{\cos(\theta)}$. $\sigma_y = 15°$ is a constant value, and $\theta \in \left[-\frac{\pi}{2}, \frac{\pi}{2}\right]$ is the latitude of the gaze point. Fig. 6 shows a qualitative comparison where the saliency maps generated from ScanTD's prediction results are better than the others. The gallery scene is from the AOI dataset [42], the room scene is from the Sailent360! dataset [32], and the building scene is from Sitzmann [35].

## 5 Conclusion and Future Work

We propose a novel method, ScanTD, which offers a potential solution for the visual task of realistic scanpaths modeling on 360° images. Through extensive experiments, we have confirmed that our method ScanTD achieves state-of-the-art performance across three test datasets. Our model, ScanTD, advances the theoretical understanding of scanpath generation in panoramic viewing and also holds significant potential for practical real-world applications. For instance, by accurately modeling gaze trajectories, our approach can aid content creators in designing VR and AR experiences that naturally guide the viewer's attention, enhancing narrative storytelling and informational clarity.

There is still much to explore in scanpath modeling, especially when considering its potential applications in various sectors. For example, the task we are currently studying is free-viewing scanpaths. However, many practical applications of VR, such as navigational training for pilots or surgical procedure simulations for medical professionals, require task-driven human visual attention data. This distinction underscores the need for developing scanpath models that are tailored to specific tasks, offering a promising research direction that could enhance the effectiveness of training and education in VR environments.

## Acknowledgments

This work was supported by Marsden Fund Council managed by the Royal Society of New Zealand under Grant MFP-20-VUW-180.

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
