# OpenReview forum: "ScanTD: 360° Scanpath Prediction based on Time-Series Diffusion"
_acmmm.org/ACMMM/2024/Conference — MM2024 Oral_

### Official Review · Reviewer_pK8u · 2024-05-20

**Rating:** 3
**Confidence:** 3

**Summary:**

This is a clearly written paper that introduces several novel methods to the domain of 360-degree panoramic images, achieving promising results in the task of Scanpath.

**Strengths:**

The main innovations of the paper are as follows:
1.	Replacing 2D CNN in ViT with Spherical CNN: This approach extracts global features from 360-degree images.
2.	Designing a Transformer-based Temporal Prediction Network: This network is used to generate gaze points.
3.	Utilizing a Conditional Diffusion Model: This model predicts points along the Scanpath.

**Limitations:**

1.	Experimental Comparisons: The results differ from the original data in the ScanDMM and ScanGAN papers when using the same metrics. The original papers averaged results over 10 runs and reported the best, whereas this paper uses 100 runs. The reasoning behind this experimental setup needs further explanation.
2.	Ablation Study Details (Section 4.4): The setup and description in the ablation study are too simplistic and lack necessary analysis. For example, in ScanTD_2, TTS was not used, so it is important to consider whether parameters were reduced in the control group. It is also worth discussing if the reduced performance can be compensated by increasing network parameters. Simply showing decreased performance without analyzing the reasons is insufficient to demonstrate the effectiveness of the innovations.
3.	Generalization Evidence (Section 4.5): Using only three saliency detection visualization examples and evaluation formulas to claim superior generalization of the model is not robust for a scientific paper. It would be better to reduce Section 4.5 and enhance the experiments and analysis in Section 4.4.
4.	Model Architecture: In the TTS model, the transition from Spatial Multi-head Attention to Temporal Multi-head Attention only changes the input feature dimension. However, the model concatenates the two feature vectors instead of running them in parallel. Would changing the order of these layers yield the same results?
5.	Feature Vector Reduction: The experimental model significantly reduces the transformer input feature vector from B * 196 * 768 to B * 20 * 196. It is necessary to evaluate whether this reduction affects the model's performance.
6.	ViT Core Concept: One of ViT's main concepts is segmenting an image into small patches and learning the self-attention between them. This experiment replaces image segmentation with a whole image, requiring further experimental validation and analysis to verify that self-attention learning is still effective.

**Suitability:**

3

---

### Official Review · Reviewer_cLKt · 2024-05-24

**Rating:** 4
**Confidence:** 3

**Summary:**

The paper proposes a novel approach called ScanTD for predicting scanpaths in 360° images. It uses a diffusion model conditioned on scene features extracted from the image by a Vision Transformer combined with Spherical CNNs. A transformer-based time-series module is incorporated into the diffusion model to capture the temporal dependencies in the scanpaths.

**Strengths:**

1. The idea of using diffusion models for scanpath prediction is novel and potentially promising.
2. The use of Vision Transformers with Spherical CNNs for extracting global scene features from 360° images is a reasonable approach.
4. The paper provides a good overview of previous work on scanpath modeling and diffusion models.

**Limitations:**

1. The paper lacks technical details and clarity in explaining the proposed method. The descriptions of the FV-Net and SD-Net components are vague, making it difficult to fully understand and reproduce the approach.
2. There is a lack of theoretical justification for the novel "temporal attention mechanism" mentioned, and no details are provided on how it works or how it improves over existing attention mechanisms.
3. The experimental results section is minimal, with only a brief statement about outperforming state-of-the-art methods on three datasets, without providing any insightful analysis.
4. More qualitative comparisons should be included. , ScanDMM looks better than the proposed method in some cases, such as the FIg. 5 (office) and Fig. 6 (Room)
5. The paper does not discuss the computational complexity or inference time of the proposed method.

**Suitability:**

2

---

### Official Review · Reviewer_zJm2 · 2024-06-03

**Rating:** 6
**Confidence:** 4

**Summary:**

Overall the paper is well written with nice results. A nice study is conducted where the proposed method ScanTD shows very promising results in terms of Scanpath prediction in 360 degree images.

The main contributions of the paper are as follows:

1) It proposes a novel 360◦ scanpath prediction method based on time-series diffusion, ScanTD, which boosts the performance of scanpath generation in 360◦ images.
2) A transformer-based time-series (TTS) module with a novel temporal attention mechanism to better capture the dynamic temporal information across time-ordered gaze points. This modification significantly improves the accuracy of both the spatial positions and the temporal order of the generated gaze points.
3) Integrates Spherical CNNs into Vision Transformer for better global scene feature extraction in 360◦ images.

**Strengths:**

The strengths of the paper are as follows:

1) The use of Vision Transformer (ViT) along with the Spherical CNN (using patch embedding) indeed helps to extract the global features from the 360 degree images.
2) A conventional U-net is replaced with a Transformer-based Time-Series (TTS) network inside the Diffusion model to process the sequential data of the diverse scanpaths.
3) The overall pipeline shows novel contribution towards Scanpath prediction.
4) The ScanTD method outperforms other methods in terms of accuracy.
5) A nice ablation study is also performed to highlight the different modules and their respective contributions to the overall pipeline.

**Limitations:**

There are some few drawbacks as mentioned below:

1) More comparisons with other state-of-the-art methods could be performed.
2) It should be mentioned explicitly, the impact of the different modules inside the pipeline based on the ablation studies.

**Suitability:**

3

---

### Meta-Review · Area_Chair_56kW · 2024-06-30

**Recommendation:** Accept (Oral)
**Confidence:** 4

**Metareview:**

This paper presents ScanTD, a novel approach for predicting scanpaths in 360° images by means of diffusion models and features extracted from the image.  All the reviewers agree on the timeliness of the approach and the good shape of the paper. Some concerns were risen during the reviewing process but the authors were able to address them in the rebuttal. Given that the authors tackle all the reviewers' comments, I believe that the paper can be accepted for oral presentation.